

# Bacterial exposure leads to variable mortality but not a measurable increase in surface antimicrobials across ant species

Omar Halawani[1], Robert R. Dunn[1], Amy M. Grunden[2] and
Adrian A. Smith[3,4]

[1] Department of Applied Ecology, North Carolina State University, Raleigh, NC, USA
[2] Department of Plant and Microbial Biology, North Carolina State University, Raleigh, NC, USA
[3] Department of Biological Sciences, North Carolina State University, Raleigh, NC, USA
[4] Research & Collections, North Carolina Museum of Natural Sciences, Raleigh, NC, USA

## ABSTRACT

Social insects have co-existed with microbial species for millions of years and have evolved a diversity of collective defenses, including the use of antimicrobials. While many studies have revealed strategies that ants use against microbial entomopathogens, and several have shown ant-produced compounds inhibit environmental bacterial growth, few studies have tested whether exposure to environmental bacteria represents a health threat to ants. We compare four ant species' responses to exposure to *Escherichia coli* and *Staphylococcus epidermidis* bacteria in order to broaden our understanding of microbial health-threats to ants and their ability to defend against them. In a first experiment, we measure worker mortality of *Solenopsis invicta*, *Brachymyrmex chinensis*, *Aphaenogaster rudis*, and *Dorymyrmex bureni* in response to exposure to *E. coli* and *S. epidermidis*. We found that exposure to *E. coli* was lethal for *S. invicta* and *D. bureni*, while all other effects of exposure were not different from experimental controls. In a second experiment, we compared the antimicrobial ability of surface extracts from bacteria-exposed and non-exposed *S. invicta* and *B. chinensis* worker ants, to see if exposure to *E. coli* or *S. epidermidis* led to an increase in antimicrobial compounds. We found no difference in the inhibitory effects from either treatment group in either species. Our results demonstrate the susceptibility to bacteria is varied across ant species. This variation may correlate with an ant species' use of surface antimicrobials, as we found significant mortality effects in species which also were producing antimicrobials. Further exploration of a wide range of both bacteria and ant species is likely to reveal unique and nuanced antimicrobial strategies and deepen our understanding of how ant societies respond to microbial health threats.

## INTRODUCTION

Organisms that live in societies, whether they are ants, termites or humans, benefit from cooperative defense, offspring care, and foraging success. However, high levels of

Corresponding authors
Omar Halawani,
oghalawani@gmail.com
Adrian A. Smith,
aasmith7@ncsu.edu

relatedness in some social species, combined with high densities of individuals, whether in big ant colonies or big human cities, can cause greater susceptibility to pathogens and parasites (*Hughes, Eilenberg & Boomsma, 2002*; *Sengupta, Chattopadhyay & Grossart, 2013*; *Schmid-Hempel, 1998*). Human societies reduce pathogenic and parasitic loads in many ways, in particular, antibiotics have saved millions of lives (*Levy, 1992*). Like humans, ants also have hygienic behaviors that reduce transmission (*Konrad et al., 2018*; *Pull et al., 2018*), can employ vaccination-like behavior (*Konrad et al., 2012*), and employ a variety of antimicrobials (*Schlüns & Crozier, 2009*). For social insects in general, and ants in particular, a first step in understanding antimicrobial strategies is to assess which microbes present health threats and how responses to microbial exposure might vary.

Extant social insect taxa have co-existed with microbial species for millions of years and have evolved a diversity of collective defenses (*Meunier, 2015*). These defenses include hygienic behaviors such as allogrooming—the grooming of nest-mates—or the removal of waste and dead individuals (*Cremer, Armitage & Schmid-Hempel, 2007*). Ants and other social insects also produce prophylactic antimicrobial compounds (*Traniello, Rosengaus & Savoie, 2002*; *Yek & Mueller, 2011*). In ants, prophylactic antimicrobial compounds are most well-documented in response to fungal entomopathogens (*Schlüns & Crozier, 2009*; *Tragust et al., 2013*). However, the microbial-rich environments of ground-nesting ants facilitates interactions with many more species of bacteria than fungi, and many of these bacteria seem likely to have the potential to kill ants, even if they are not specialized entomopathogens (*Hoggard et al., 2013*; *Ishak et al., 2011*).

How harmful exposure to non-entomopathogenic bacteria is to ants is largely unknown. However, many studies have demonstrated that compounds produced by ants are effective at inhibiting growth of non-entomopathogens. For example, one early study showed inhibition of the gram-negative bacterium *E. coli* (*Mackintosh et al., 1995*) by peptides synthesized by the metapleural glands of the ant *Myrmecia gulosa* (Fabricius, 1775). Concentrations of metplueral gland secretions of at 10-ant equivalents had a 99% kill rate on microbial cells. Metaplueral gland peptides can disrupt the cell walls of gram-negative bacteria (*E. coli*) as well as plasma membranes of gram-positive bacteria (*Bacillus cereus*) (*Mackintosh et al., 1995*). The venom from *Solenopsis invicta* (Buren, 1972) contains multiple alkaloids that have different levels of antimicrobial effectiveness against gram-negative and gram-positive bacteria (*Blum et al., 1958*), with a generally higher ability to inhibit gram-positive bacteria (*Jouvenaz, Blum & MacConnell, 1972*). In another study, *S. invicta* venom alkaloids were isolated and found to be effective at inhibiting *Pseudomonas fluorescens* biofilms (*Carvalho et al., 2019*). Finally, a comparison of surface extracts from 20 ant species from four subfamilies found that extracts from 40% of tested species were ineffective at inhibiting the growth of *Staphylococcus epidermidis* at just a 5-ant extract equivalent (*Penick et al., 2018*). That comparative study as well as others on wasps (*Hoggard et al., 2011*, *2013*), bees (*Stow et al., 2007*), and thrips (*Turnbull et al., 2011*) draw evolutionary conclusions about these organisms' antimicrobial activity by assessing their ability to inhibit a non-entomopathogenic bacterial species (usually *Staphylococcus spp.*). However, although ant-produced compounds are capable of inhibiting these bacteria, we do not know whether exposure to
non-entomopathogenic and unfamiliar bacteria represent an actual health threat that ants would benefit from being able to defend against.

Exposure to pathogens often produces physiological immune responses, where low-risk challenges of pathogens upregulate individual and group immunity. In a study with the ant *Formica exsecta* Nylander, 1846, workers were orally exposed to *Serratia marcescens, E. coli*, or *Pseudomonas entomophila* (*Stucki et al., 2019*). Antimicrobial gene expression of ants exposed to *P. entomophila* and *E. coli* was upregulated after exposure, possibly as a general immune response from exposure to a high bacterial load or as the ant's prophylactic response. The response of *F. exsecta* suggested that even non-entomopathogenic bacteria (*E. coli*) can provoke immune or antimicrobial responses from ants. However, extrapolation of results from a few species of ants to general patterns are likely to mislead, as ants are a diverse group of organisms with variable life history traits that likely result in different strategies for dealing with microbial exposure. Comparative studies across ant species are likely to reveal key differences in antimicrobial strategies.

Recently, the lethality of fungal pathogen exposure was compared across 12 species of ants from 4 genera (*Bos et al., 2019*). The study found species-level mortality differed across, and even within, ant genera in response to identical pathogen exposure. In addition to this, the three mostly closely related species tested had the most similar responses, suggesting disease resistance might be phylogenetically linked traits. Similar conclusions were reached in a study of surface antimicrobials collected from 20 species of ants (*Penick et al., 2018*). The antimicrobial ability, measured through inhibition of *S. epidermidis* cultures, using ant-derived surface extracts was highly variable across species; however, there was a phylogenetic signal associated with species that yielded inhibitory extracts.

In this study, we consider four ant species (*Solenopsis invicta, Brachyponera chinensis* (Emery, 1895), *Aphaenogaster rudis* Enzmann, 1947, and *Dorymyrmex bureni* (Trager, 1988)) whose surface antimicrobials have been shown to be differently effective at inhibiting the human-associated bacteria species, *Staphylococcus epidermidis* (*Penick et al., 2018*). In *Penick et al. (2018)*, surface extracts from *S. invicta* and *D. bureni* workers were very effective at inhibiting growth of *S. epidermidis*, whereas *A. rudis* and *B. chinensis* ants showed weak inhibitory ability. These results and the above referenced pharmacological work indicating that some ant-derived compounds effectively inhibit the growth of *S. epidermidis* and *E. coli* bacteria have led us to the research question of whether non-ant associated bacteria represent an actual health threat to different ant species. Perhaps ants which are susceptible to exposure to a broad range of bacteria produce broadly effective antimicrobials as a compensatory strategy. Therefore, we first compare the lethality of bacterial exposure across those four ant species to a gram-positive bacterium, *S. epidermidis*, and a gram-negative bacterium, *E. coli*. We do this by exposing live groups of ant workers with agar-grown bacteria and measuring their mortality over 48 h.

In a second experiment, we tested if exposure to bacteria under these conditions resulted in an increase in antimicrobial compounds on the bodies of two of these species (*S. invicta* and *B. chinensis*). In a previous study, surface extracts from *S. invicta* were

strong inhibitors of bacterial growth, and *B. chinensis* lacked inhibitory ability (*Penick et al., 2018*). One possible explanation for this difference is that some species may deploy antimicrobial compounds only in response to exposure to harmful microbes. We test this by extracting surface compounds from two species of bacteria-exposed ants and testing their bacterial inhibitory ability as compared to a control. We predicted that exposure to a potentially lethal bacterial challenge would result in an increase of antimicrobial ability from ant surface extracts, which would indicate that the antimicrobial abilities of some ants are able to be conditionally deployed to meet direct microbial challenges.

## MATERIALS AND METHODS

Bacteria were sourced from Carolina Biological Supply Company. *Staphylococcus epidermidis* and *Escherichia coli* strain K12 were kept in glycerol stocks at −80 °C, then spread onto BD Difco™ LB, Miller (Fisher Scientific) agar 100 mm × 15 mm petri dish plates. Plates were incubated for 24 h at 35 °C. Bacteria were cultured on plates for experiment 1 at 0.5 MacFarland standard. For experiment 2, bacteria from agar plates were also cultured in liquid BD Difco™ LB, Miller media. Liquid cultures were incubated for 24 h at 35 °C before being used in assays.

### Experiment 1: Ant mortality in response to exposure to bacteria

Worker mortality in response to bacterial exposure was measured across four ant species. Fire ants, *Solenopsis invicta*, were collected between June and December of 2018 in Raleigh, NC, USA by collecting the tops of mounds with a shovel. *Aphaenogaster rudis* group ants were collected in Durham, NC, USA and Raleigh, NC from July to August of 2018 by aspirating workers from within their nests. Asian needle ants, *Brachyponera chinensis*, were collected in Raleigh, NC in December 2018 to May 2019 by opening dead logs to expose workers for collection via aspiration. *Dorymyrmex bureni* colonies were collected in Hoffman, NC, USA in June 2019 by collecting mounds with a shovel. Permitted access to field collection sites was provided by North Carolina Wildlife Resources Commission and Duke University, Office of the Duke Forest. Species used in this and the following experiment were identified using morphological keys (*MacGowen, 2014*), and voucher specimens are in the collection of author AAS at the North Carolina Museum of Natural Sciences. Each species of ants was separated from its nesting material to reduce further environmental (microbial) interaction and kept in the lab for <24 h before experimentation. Before experimentation, while in the lab, ants were given a supply of 20% sugar water (*Kay et al., 2014*).

In each exposure treatment, we used 200 *S. invicta* workers and 50 *B. chinensis*, *D. bureni*, and *A. rudis* workers (Fig. S1). The difference in number of workers used for *S. invicta* was based on what could be consistently captured from colonies. Experiments with each species were replicated across 15 colonies, where each colony was subject to three treatments. Treatments were: exposure to *E. coli*, exposure to *S. epidermidis*, or exposure to sterile agar as control. Exposure experiments were performed in lid-covered deep petri dishes (100 mm × 25 mm). The dishes were prepared with Insect-A-Slip

(BioQuip Products, Compton, CA, USA) applied around the interior edge to prevent ant escape. Ants were exposed to a small piece of agar (10 mm × 9.7cm$^2$) that had a bacterial lawn of *S. epidermidis*, *E. coli*, or no bacterial growth as a control. Bacterial lawns were grown from liquid culture incubated overnight at 35 °C in a shaking incubator adjusted to a 0.5 MacFarland standard. In the experimental arena, agar pieces with and without bacteria were hydrated with 100 μl of MilliQ H$_2$O, this prevented agar from drying out and encouraged ant interaction with agar. Depending on the species being tested, all 200 or 50 ant workers were introduced at once to the experimental arena after water was absorbed by agar by directly placing ants on agar to ensure at least one point of direct exposure. Mortality was measured by counting dead ants at 24 h and 48 h. Observations were recorded when ants tunneled into agar, as an additional indication of direct interaction with the bacteria.

We tested the mortality effect of exposing a group of workers as compared to workers from the same colony exposed to control, non-bacterial, treatments. Data at 24 h and 48 h were graphed and analyzed with a Friedman's rank sum test with Finner post hoc pairwise comparison. Friedman's test is similar to a one-way ANOVA for an unreplicated complete blocked design of nonparametric data (*García et al., 2010*). The Finner post hoc comparison procedure was used for indicating significance of treatment effects. Finner's post hoc uses a step-down *p*-value adjustment value; it rejects test statistics when $p_i > 1 - (1 - a)^{(k-1)/I}$ (*García et al., 2010*). Friedman's statistical tests used at a significance level of α = 0.05, and Finner post hoc used an adjusted threshold level of significance of $\alpha_1 = 0.0975$, $\alpha_2 = 0.05$, or $a_3 = 0.0336$. Analysis was carried out in R version 3.5.2 (*R Core Team, 2017*) using the packages scmamp (*Calvo & Santafe, 2016*) and devtools (*R Core Team, 2017*).

## Experiment 2: Does exposure to bacteria lead to an increased presence of surface antimicrobials?

We compared the antimicrobial ability of surface extracts from bacteria-exposed and non-exposed *S. invicta* and *B. chinensis* worker ants. We wanted to know if antimicrobial activity of surface extracts was a constant feature of these species, or something that might be responsive and increasing when exposed to bacterial threats. For the experiment, we collected *B. chinensis* ants by opening wood logs with their nests and aspirating workers. The ants were collected between January and May 2019. To collect *S. invicta* ants, the tops of nest mounds were removed, and workers were aspirated from removed soil. Ants were collected between June and December 2018. Both species were collected in Raleigh, NC, USA.

The experiment consisted of an exposure treatment followed by extracting surface compounds of workers to be challenged by bacteria in a liquid culture well plate assay (Fig. S2). As was done for experiment 1,200 *S. invicta* workers and 50 *B. chinensis* workers were used for each exposure treatment. We used 15 colonies per species for treatments, in an incomplete block design. Treatments were exposure to *S. epidermidis*, *E. coli*, or sterile agar for control, following the protocol used for experiment 1 described above. Bacterial lawns were grown from liquid bacterial cultures of *E. coli* and *S. epidermidis*

adjusted to a 0.5 MacFarland standard after growing overnight at 35 °C in a shaking incubator. A total of 100 μl of MilliQ H2O was pipetted under the agar piece to hydrate the media and encourage ants to interact with the agar plugs. Exposure treatments ran for 6 h. Then, any dead individuals were removed from exposure before freeze-killing the remaining living workers for extraction. The 6 h exposure time was determined from earlier trials and selected as a, largely, sub-lethal level of exposure as lethal effect of exposure were first seen after 6 h.

Surface compounds of 40 workers from each exposure treatment were extracted in 360 μl isopropanol for 15 min after an initial vortex spin for 15 s. Whole workers were used, and ant bodies remained in-tact through the vortexing and solvent extraction. Previous studies have shown effectiveness of using polar solvents (e.g., ethanol) to extract antimicrobial compounds (Penick et al., 2018; Stow et al., 2007; Turnbull et al., 2011). Extracts were filtered through 0.2 micron SEP filters (Stow et al., 2007). Isopropanol extracts were evaporated in a vacuum centrifuge then resuspended in 360 μl LB media.

We adjusted a 96 well plate assay protocol for testing antimicrobial ability against bacterial liquid culture (Stow et al., 2007). The initial 40-worker-ant extract was divided into 4 equal parts; 10 ant equivalents of ant body-surface extracts (90 μl) were tested against 100 CFU's of either *S. epidermidis* or *E. coli*. Bacterial controls of 100 CFU's bacteria (*S. epidermidis* or *E. coli*) with LB media were plated as maximum growth controls. Another 10 ant equivalent extracts suspended in LB media were plated as minimum growth controls. Media controls were also plated to verify there was no contamination. Plates were incubated at 35 °C for 18 h before pipetting 10 μl of WST-8 (PromoKine, Heidelberg, Germany) followed by another hour of incubation to allow the salt to bind to available live cells. WST-8 salt binds to cellular membranes of organisms that are undergoing active transport, and therefore, provides a colorimetric method to analyze live microbial count in culture (Braun et al., 2018). After incubation, optical density was measured at $OD_{600}$ with 5 s shaking before reading using a plate reader (SpectraMax M5; Molecular Devices, San Jose, CA, USA).

$OD_{600}$ readings of the well plate were comparable values across the plate because of WST-8 binding equally to available live cells (PromoKine, Heidelberg, Germany). Readings of exposure treatment workers were measured as percent inhibition adjusted relative to the control treatment workers. Antimicrobial ability was determined by comparing adjusted percent inhibition to plated maximum and minimum growth controls. Final results were reported as change in percentage of inhibition of microbes between treatment and control groups. Results provided either a positive or negative inhibition value based on the treatment inhibition relative to the control.

In our results some of the outlying data points are percent inhibition measures beyond 100% and below 0%. Each data point is a measure of inhibition in wells where half of the ant extract is added to an experimental well with live bacteria; however, we report this measure relative to two other wells: one containing only bacteria and media (a maximum bacterial growth control) and the second containing only media and the other half of the ant extract (a minimum growth control). For the outlying points beyond 100%, the half of the ant extract used as minimum growth control had a higher absorbency reading

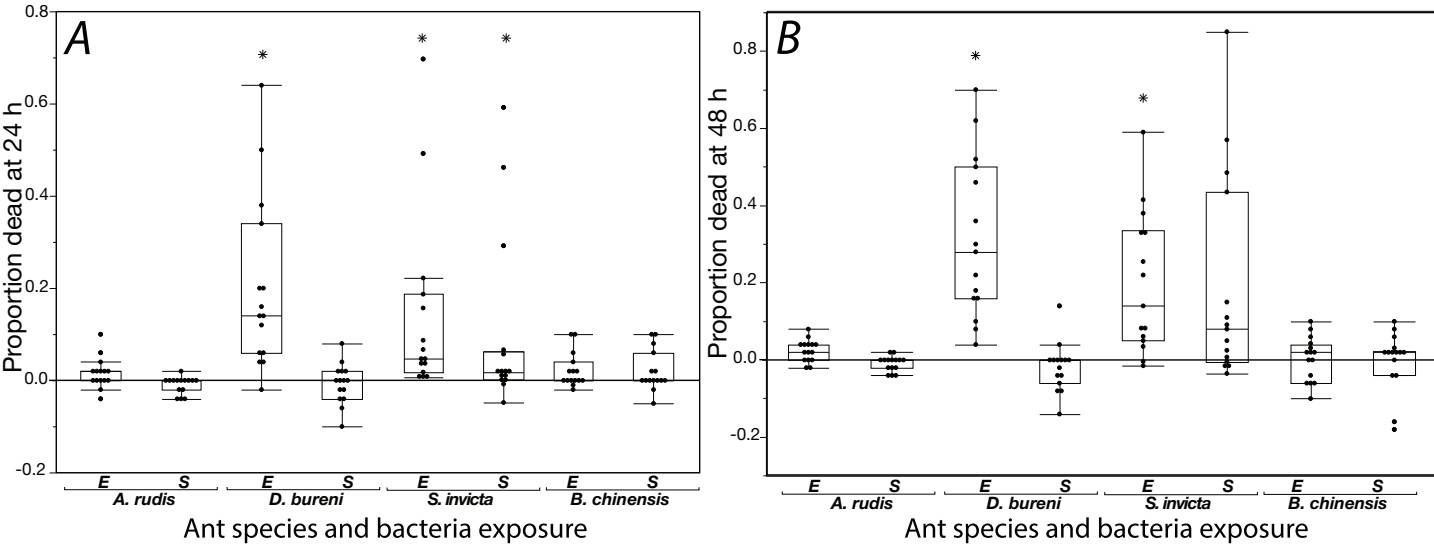

**Figure 1** **Mortality is shown for data collected at 24 h (A) and 48 h (B) when ants were exposed to *S. epidermidis* (S) or *E. coli* (E) after adjusting treatment values to control value.** A total of 15 colonies were tested for each species. Asterisks indicate statistically significant difference in mortality, relative to the control according to Friedman's rank sum test and Finner post-hoc test. Box plot line represents median values, whiskers represent 1st and 3rd quartile, data points represent ant colonies, outliers are data outside of 1.5 IQR.

(the proxy for bacterial growth) than the other half of the ant extract used in the experimental well. Minimum growth control wells could have higher absorbency readings from WST-8 binding to available molecules present in the extract, such as carbohydrates or even secondary products that could be contributing to the antimicrobial ability seen in the experimental well. For the outlying points below 0%, the experimental well absorbency was higher than the maximum growth control. This was likely due to the ant extract failing to inhibit the microbial challenge in the experimental well, and instead supplementing the growth of the bacteria culture.

We tested if antimicrobial ability showed a conditional response with a Kruskal Wallis rank sum test. The Kruskal Wallis analysis tests non-parametric data for multiple groups when assumptions for ANOVA are not met—in our case, our data was not a complete design. Some replicates were comprised of samples from two nests to complete all three treatments. In these instances, a control and single treatment group were sampled from colonies twice—once for *E. coli* and once for *S. epidermidis*. All statistical tests used a threshold level of significance of $\alpha = 0.05$. Analysis was carried out in R version 3.5.2 (*R Core Team, 2017*) using the packages dplyr (*Wickham et al., 2018*) and devtools (*R Core Team, 2017*).

## RESULTS

### Experiment 1: Ant mortality in response to exposure to non-entomopathogenic bacteria

The effects of exposure for all ants to *S. epidermidis* and *E. coli* after 24 h and at 48 h are shown in Fig. 1. In control groups across our experiment, mortality was low with average

**Table 1 Statistical comparisons for experiment 1, corresponding with Fig. 1.**

| Species | Friedman's chi-squared | df | p value | Post-hoc comparisons: Finner p value; Cohen's d effect size | | |
| --- | --- | --- | --- | --- | --- | --- |
| | | | | E. coli vs. control | S. epidermidis vs. control | S. epidermidis vs. E. coli |
| **24 h exposure** | | | | | | |
| *S. invicta* | 8.93 | 2 | p = 0.011 | **0.0001; 0.74** | **0.0026; 0.68** | 0.41; 0.18 |
| *D. bureni* | 23.33 | 2 | p < 0.0001 | **0.0005; 1.27** | 0.78; 0.16 | **0.0003; 1.32** |
| *A. rudis* | 6.63 | 2 | p = 0.036 | 0.23; 0.56 | 0.37; 0.53 | 0.062; 0.92 |
| *B. chinensis* | 2.23 | 2 | p = 0.33 | | | |
| **48 h exposure** | | | | | | |
| *S. invicta* | 16.3 | 2 | p = 0.022 | **0.021; 0.8** | 0.087; 0.69 | 0.42; 0.04 |
| *D. bureni* | 19.733 | 2 | p < 0.0001 | **0.0005; 5.74** | 0.42; 0.34 | **0.00003; 5.95** |
| *A. rudis* | 10.03 | 2 | p = 0.0066 | 0.07; 0.77 | 0.37; 0.38 | **0.011; 1.11** |
| *B. chinensis* | 2.84 | 2 | p = 0.24 | | | |

Note:
  Bold entries correspond with statistical significance in post-hoc analyses.

proportions of 0.027 at 24 h and 0.064 at 48 h. In addition, across all treatments and all species tested, agar discs had visible evidence of excavation upon first inspection when the 24 h mortality count was made.

*Solenopsis invicta* showed significant mortality to exposure (Table 1; Fig. 1). At 24 h, there was significant mortality from *S. epidermidis* and *E. coli* (Friedman's ANOVA $F(2) = 8.93$, $p = 0.011$; post-hoc analyses Finner $p < 0.01$, Cohen's $d$ effect sizes $\geq 0.68$). At 48 h, there was only significance from exposure to *E. coli* (Friedman's ANOVA 48 h $F(2) = 16.3$, $p = 0.022$; post-hoc analyses Finner $p = 0.003$, Cohen's $d$ effect size 0.8). *Dorymyrmex bureni* also displayed significant mortality to *E. coli* exposure at 24 h (Friedman's ANOVA 24 h $F(2) = 23.3$, $p < 0.0001$; Finner $p < 0.001$, Cohen's $d$ effect size 1.27) and 48 h (Friedman's ANOVA 48 h $F(2) = 19.7$, $p < 0.0001$; Finner $p < 0.001$, Cohen's $d$ effect size 5.74; Fig. 1). A significant treatment effect was detected for *A. rudis* at both 24 and 48 h, but in post-hoc comparison the only significant difference between groups was comparing mortality bacterial treatments at 48 h (Friedman's ANOVA 48 h $F(2) = 10.3$, $p < 0.01$; Finner $p = 0.01$, Cohen's $d$ effect size 1.11). All other comparisons were not significant, and mortality from both bacterial treatments was not different from the control. *Brachyponera chinensis* showed no significant mortality to either treatment (Table 1).

## Experiment 2: Does exposure to bacteria lead to an increased presence of surface antimicrobials?

Exposure seemed to have no effect on surface antimicrobial ability for both *S. invicta* and *B. chinensis*. Antimicrobial ability was calculated from $OD_{600}$ readings after WST-8 was added to wells and plates were incubated for one hour. Figure 2 shows *S. invicta* has similar antimicrobial ability to inhibit *E. coli* (Fig. 2A) or *S. epidermidis* (Fig. 2B) with and without treatment exposures, *E. coli* (KW chi-squared = 0.36, df = 1, $p = 0.55$,

Peer J

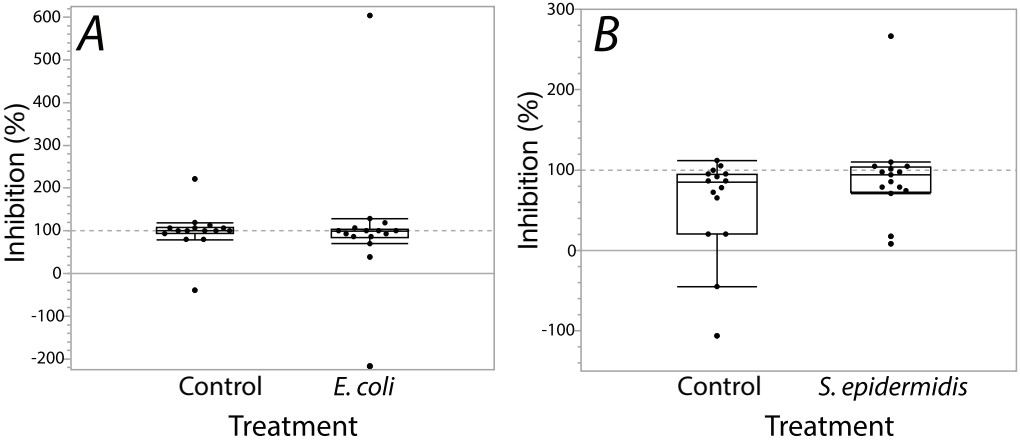

**Figure 2 Extracts from *S. invicta* effectively inhibited bacterial challenge similarly with or without treatment exposures.** Bacterial inhibition of isopropanol extracts from ants when exposed to either *E. coli* (A) or *S. epidermidis* (B) or control agar (A and B). Values are percent inhibition of extracts compared to growth controls. There are no statistically significant differences between control and treatment groups. Box plot line represents median values, whiskers represent 1st and 3rd quartile, data points represent ant colonies, outliers are data outside of 1.5 IQR.

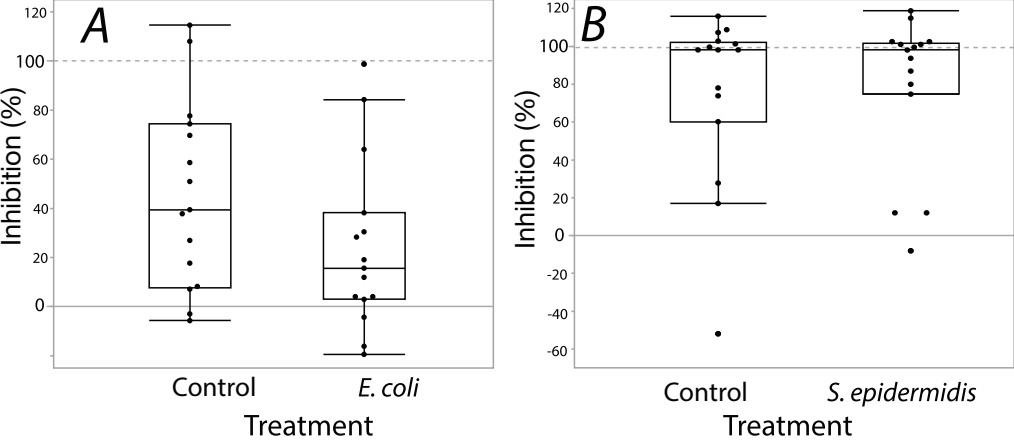

**Figure 3 Extracts from *B. chinensis* showed similar bacterial inhibition ability with or without treatment exposures.** *B. chinensis* inhibition potential for control and treatment exposures for isopropanol extracts when exposed to *E. coli* (A) or *S. epidermidis* (B). Values are percent inhibition of extracts compared to growth controls. There are no statistically significant differences between control and treatment groups. Maximum and minimum growth controls visualized at 100% and 0%, respectfully.

epsilon square effect size 0.01), *S. epidermidis* (KW chi-squared = 1.3011, df = 1, $p$ = 0.254, epsilon square effect size 0.045). Figure 3A shows *B. chinensis* lacks ability to completely inhibit *E. coli* but has antimicrobial ability to inhibit *S. epidermidis* (Fig. 3B). Exposure to bacteria did not lead to significant changes in antimicrobial ability (*S. epidermidis* KW chi-squared = 0.01, df = 1, $p$ = 0.92, epsilon square effect size 0; *E. coli* KW chi-squared = 2.42, df = 1, $p$ = 0.12, epsilon square effect size 0.08).

## DISCUSSION

Our first experiment demonstrated that exposure to a common bacterium—*E. coli*—could kill two of the four ant species we tested, *S. invicta* and *D. bureni*—two species that displayed strong antimicrobial ability (*Penick et al., 2018*). Exposure to *S. epidermidis* had no significant effect on any of the tested ants except for *S. invicta* workers at 24 h of exposure. The other two ants we tested, *B. chinensis* and *A. rudis* group, did not previously demonstrate antimicrobial ability nor did they die from exposure to either bacterium used in this study. Our finding indicates that, like antimicrobial ability, susceptibility to bacterial exposure varies across species. Additionally, our results could be an indication of a linkage between an ant species' susceptibility to bacterial exposure and antimicrobial ability. However, this requires further testing and considerations that our experiments did not account for, as will be discussed below.

In our second experiment, *S. invicta* and *B. chinensis* were subjected to a sub-lethal period of bacterial exposure followed by an antimicrobial assay of their surface extracts. We aimed to test for a possible increase in antimicrobial ability in response to exposure to the focal bacteria species. We predicted that exposure to a bacterial challenge would result in an increase of antimicrobial ability from ant surface extracts, which would indicate that surface antimicrobials of some ants are able to be conditionally deployed to meet direct microbial challenges. As tested, we found no evidence of a change in antimicrobial ability akin to any response to bacterial exposure. Fire ant, *S. invicta*, extracts demonstrated near-complete inhibitory ability, as was previously found (*Penick et al., 2018*), against *S. epidermidis* independent of previous exposure to the bacteria species. In addition, *E.coli* growth was inhibited by the surface extracts of fire ants. Surface extracts from *B. chinensis* also inhibited growth of *S. epidermidis*. This differs from the result of *Penick et al. (2018)* which found no effect of surface extracts of *B. chinensis* on *S. epidermidis*. Our study used isopropanol as a solvent for extraction. In contrast, *Penick et al. (2018)* used ethanol. Surface antimicrobials on *B. chinensis* may be non-polar antimicrobial compounds that ethanol could not extract.

Our experiments focused on how ants respond to exposure to bacterial species that are not known to have an evolved relationship with the ants. Our aim was to broaden our understanding of how microbes might be influencing antimicrobial traits of ants. Many studies of bacteria in ants focus on bacterial species that have specific, and often co-evolutionary, associations with insects. For instance, intracellular bacteria that have evolved to be largely maternally transmitted. *Wolbachia* species are commonly associated with insects, including ants (*Ramalho, Bueno & Moreau, 2017*). The bacteria likely target reproductive function but do not seem to kill infected adult ants (*Wenseleers et al., 1998*; *Kautz, Rubin & Moreau, 2013*). Species of the genus *Spiroplasma* were found in many of the 95 ant genera considered in a recent study (*Kautz, Rubin & Moreau, 2013*). *Spiroplasma* species can be pathogens of both vertebrates and plants; *Kautz, Rubin & Moreau (2013)* hypothesize that they might also negatively affect ants, but to date this hypothesis remains untested. In another study a *Pseudomonas* species was found by *Lofgren, Banks & Glancey (1975)* on dead ant bodies. The authors suggested that this

species might have killed the ants. The health effects that internal and environmental bacteria have on ants are only beginning to be explored.

Our experiments involved exposure to extremely high concentrations of bacteria (full "lawns" of bacterial growth on agar). These high concentrations are likely beyond what ants normally encounter. In our experiments bacterial exposure was lethal for antimicrobial producing ants, however under more natural concentrations the antimicrobial tactics the ants use might save them from the mortality effects seen in our experiments. Secondly, our experiments isolated workers outside of their normal nesting environments. These non-natural situations were used in this first experiment as a means of assuring the ants interacted with and were exposed to the bacteria. However, nesting environment and behavioral responses such as isolation and avoidance of bacterial-rich areas could be integral antibacterial strategies that ants use, which were not available to them in our study. Certainly, the microbiomes that ants maintain in their natural nests are known to be distinct from the surrounding environment and low in number potential bacterial and fungal pathogens (*Lucas et al., 2019*). In addition, though evidence of the ants excavating the agar was observed, we did not assess the number of bacteria ingested by the ants. Perhaps species that showed low mortality herein are species that ingested few or no bacteria, and species that showed high mortality ingested more bacteria. Further experiments that directly assess the health impacts of orally ingesting these non-ant associated bacteria would likely lead to interesting insights on ant immune and antimicrobial responses.

Our results hint at an association between ants' use of surface antimicrobials and susceptibility to bacterial exposure. We hypothesize that ants that are susceptible to a broad range of bacteria are more likely to employ antimicrobials (and perhaps less likely to rely on immune responses). This hypothesis could be further tested through the comparison of the antimicrobial responses of a larger number of ant species, perhaps in concert with immunological assays.

Although we were unable to demonstrate that exposure altered ants antimicrobial abilities, other studies have shown that ants can immediately change their grooming rates (*Reber et al., 2011*) and antimicrobial venom usage when threatened by pathogens (*Graystock & Hughes, 2011*). Metapleural gland secretions in *Acromyrmex octospinosus* (Reich, 1793) leaf-cutting ants were found to be significantly greater 12 h after exposure to some fungal pathogens (*Yek et al., 2012*). It is possible that our 6 h exposure was not long enough for the ants to change metapleural gland expression, or that our exposure treatments were beyond a level that would induce a measurable response. In addition, our preparation of solvent extracts using isopropanol may not have enabled collection of some active compounds which the ants might have been producing in response to our bacterial treatments.

## CONCLUSIONS

Ants, like humans, have evolved in the context of diverse bacterial communities for millions of years. However, unlike humans, "ants" represent >16,000 species each with

unique life-histories, adaptations, and potential antimicrobial strategies. Like other recent comparative studies in ants (*Bos et al., 2019*; *Penick et al., 2018*), our results show that responses to potentially harmful microbes are varied among ant species. To ants that are susceptible to microbial exposure, threats are not limited to entomopathogenic bacteria. Previous pharmacological studies have demonstrated that some ants produce compounds that inhibit non-ant-associated bacteria (*Carvalho et al., 2019*; *Jouvenaz, Blum & MacConnell, 1972*). These studies may have been revealing that those ants are susceptible to and thus are defending against exposure environmental bacteria. Further exploration of a wide range of both bacteria and ant species is likely to reveal unique and nuanced antimicrobial strategies. Continued research is likely to deepen our understanding of how ant societies maintain health and how we might adopt some of these non-human-society strategies to better improve our own responses to microbial health threats.

## ACKNOWLEDGEMENTS
We thank Clint Penick and Stephanie Mathews for their guidance on our methods and analysis.

### Funding
Funding for this study was provided by a Triangle Center for Evolutionary Medicine (TriCEM) grant awarded to Adrian Smith. There was no additional external funding received for this study. The funders had no role in study design, data collection and analysis, decision to publish, or preparation of the manuscript.

### Grant Disclosures
The following grant information was disclosed by the authors:
Triangle Center for Evolutionary Medicine (TriCEM).

### Competing Interests
The authors declare that they have no competing interests.

### Author Contributions

- Omar Halawani conceived and designed the experiments, performed the experiments, analyzed the data, prepared figures and/or tables, authored or reviewed drafts of the paper, and approved the final draft.
- Robert R. Dunn conceived and designed the experiments, authored or reviewed drafts of the paper, and approved the final draft.
- Amy M. Grunden conceived and designed the experiments, authored or reviewed drafts of the paper, and approved the final draft.
- Adrian A. Smith conceived and designed the experiments, prepared figures and/or tables, authored or reviewed drafts of the paper, and approved the final draft.

## Field Study Permissions

The following information was supplied relating to field study approvals (i.e., approving body and any reference numbers):

Field permit was provided by North Carolina Wildlife Resources Commission (#17-SC01086) and the Duke Forest Teaching & Research Laboratory (#R1617-437).

## Data Availability

The raw data for Figs. 1–3 and the data analysis code used are all available in the Supplemental Files.

## Supplemental Information

Supplemental information for this article can be found online at http://dx.doi.org/10.7717/peerj.10412#supplemental-information.

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
