# Peer review of "Bacterial exposure leads to variable mortality but not a measurable increase in surface antimicrobials across ant species"

_PeerJ, doi:10.7717/peerj.10412_

## Round 0.1 · original submission · Major Revisions

· Academic Editor

Major Revisions

This manuscript has now been reviewed by three experts in the field, and all three have made thoughtful and detailed comments on the manuscript. They identified some issues with the direction and organization of the introduction and discussion. Please respond to each point raised in a rebuttal letter and indicate where and what changes have been made in the manuscript. Thank you, and I'm looking forward to reading your revised manuscript.

Reviewer 1 ·

Basic reporting

Dear editor and author,
Thank you very much for the invitation; it was a pleasure to do the manuscript review. The study titled “Lethal and antimicrobial responses to bacterial exposure across ant species” it aims to compare four ant species’ responses to non-pathogenic bacterial exposure in order to broaden the understanding of microbial health-threats to ants and their ability to defend against them. The results are interesting and after some modifications it could be considered for publication.

Here is my main concern:

-Authors should consider restructuring the entire introduction. The introduction is long and has no objectivity. In addition, the four species used in the study were not highlighted in the introduction and it would be of great help to the reader if in the introduction the defense strategies against pathogens of each species were presented.

-Adding figures with experimental design of the two experiments would be of great help to the reader.

- Authors should consider presenting a mortality curve for each species and treatment to illustrate and highlight the main findings of the study.

Specific comments:
L1: Title is not very informative. I suggest focusing on the main results.

L22: It would be interesting for the reader to know here what these four species are.

L44: Please provide references.

L48-50: Please provide references and examples.

L51-53: It seems strange to me: using ant antibiotics to understand which microbes are pathogenic. How is this possible? I believe that a study showing the function of these bacteria would be more efficient to discover this. Could you provide more details?

L68: It would be of great to give credit to the taxonomist for all the ant species you mention in the text. Check this out on antcat.org

L106: What are the species? Restructuring the introduction would be of great help to the reader. Provide information on each of the species and strategies that each would have the greatest impact on the study.

L129: You need to assume that readers are not American and do not know what this location. I suggest adding the country and the geographic coordinates here.

L129: Finally, the 4 species! Authors should reveal these species in the abstract, in the introduction, and provide data on the strategies for defending these species in the introduction.

L129-135: How were these species identified? Morphology? Which taxonomic key? Were these ants deposited in an entomological collection? Do you have a voucher?

L136-138: This information should be moved for introduction and not in MM.

L139-149: it would be very helpful for the reader to explain why this.

L142: How long did these ants stay in the lab until the experiment was carried out. I wonder if stress is a factor to be considered as responsible for the results of the study.

L155: How many ants were added in each treatment? Were they together?

L157: How many ants died in the control group? This will tell me if the ants were under stress.

L175: Why only those two? Do the other species not produce inhibitors? A restructured introduction should address these issues.

L175-177: This information should be moved and explored in the introduction.

L183: Same thing about the experiment 1. For the manuscript to gain international appeal, the authors should also offer the country and the geographic coordinates.
How were these species identified? Was it deposited in an entomological collection? Please provide vouchers.

L185: 200 in total, right? and how many were added in each treatment? It is still difficult to visualize and understand the experiments. I believe that figures with experimental designs could offer more support to the reader.

L194-195: Do you have data that provide a basis for this choice? These data would give more credibility to this choice and to the study in general. Please provide this data.

L234-237: Excellent! It would be interesting to put the n of dead ants in control. In addition, providing a mortality curve for each treatment and each species would be very helpful and would help to highlight the findings of the present study.

L242-243: Where is this result? Quote the Figure or Table here.

L285: Please, remove that square.

L350-352: Please provide references.


Figure 1: Very small and poor-quality image.
It would be interesting if the authors showed the mortality curve of this first experiment with respect to the control group.
If I managed to visualize the graph well, A rudis and B. chinensis was no different from the control group in terms of mortality, is that it? Can we say that these ants were under stress and can we not consider these results, whether positive or negative?

Figure 2 and 3: It would gain more impact if the authors added the statistic to that image (p-value).

Experimental design

It has already been commented above.

Validity of the findings

It has already been commented above.

Additional comments

It has already been commented above.

Reviewer 2 ·

Basic reporting

In the manuscript titled „Lethal and antimicrobial responses to bacterial exposure across ant species” the authors investigate a whether surface substances ants influence the ant’s ability to survive exposure to two bacteria that likely did not coevolve with the ants and whether the antimicrobial activity of the ant’s surface substances changes upon exposure to these bacteria. Unfortunately, although I liked the basic premise of their two experiments, I cannot recommend the study in the present format as it contains some lines of argumentation that I think the authors should reconsider, methods that might be inappropriate to answer the questions raised by the authors and at places a poor choice of references.

Experimental design

See 3. Validity of the findings

Validity of the findings

1. Argumentation.
I like the authors idea to use bacterial species that likely did not coevolve with the ants, as I agree with the authors statement that harmfulness of bacteria or other microorganisms is often only tested with likely pathogens. However, exactly this point is also problematic and I think the authors need to reconsider their lines of argumentation for the following points.
1.1 The authors may find one effect with the bacteria they used but potentially an opposite effect with other bacteria. As there are literally tons of different bacteria in the environment of ants, choice of representative bacteria becomes crucial. As far as I understand this choice was mainly based on a previous study by Penick et al (2018) and I wonder why the authors did not choose for example widespread bacteria species that these ants are likely to encounter in their environment. Moreover, in line 82-84 the authors write: “we do not know whether exposure to these non-entomopathogenic bacteria represents an actual health threat that ants would benefit from having evolved defences against.” This implies that part of the aim of the authors study was to investigate evolved defences, i.e. antimicrobial surface substances. Altogether, this leaves the reader wondering what exactly the expectation of the authors was investigating evolved defences against non-coevolved bacteria. That susceptibility to bacteria varies under these premises (main finding reported in line 289) is hardly surprising.
1.2 In line 309-310 the authors state that their aim was to broaden our understanding of how microbes might influence antimicrobial traits of ants. Have the authors considered that antimicrobial traits might also influence microbes, especially those in the environment of ants?
1.3 At the end of the discussion the authors raise the hypothesis (line 332-333) that ants that are susceptible to environmental bacteria are more likely to employ antimicrobials. This brings me to the part of the study that I consider most problematic: In contrast to for example fungal entomopathogens that enter their host via penetration, bacterial entomopathogens and likely also the non-entomopathogenic bacteria used in this study usually represent a health threat to insects when orally ingested. However, once ingested surface substances are unlikely to play a role in the fight against bacteria. How this might influence the results presented in the study or the hypothesis raised in line 332-333 is unfortunately completely ignored.
1.4 Another argumentation line that should be reconsidered is in line 310-319: here the authors criticize the use of bacteria that show a coevolutionary association with insects. However while the authors used in their experiment free-living bacteria, examples given here concern the intracellular bacteria Wolbachia and Spiroplasma which, apart from several other differences, cannot be compared as they are most often maternally transmitted and will likely never encounter surface substances of ants.

2. Methods.
2.1 In their first experiment on the mortality of ants upon exposure to bacteria on an agar block the authors did not control exposure. Although they write in line 156 that they ensured direct contact between ants and bacteria at least once and in line 157-158 that observations of ants tunnelling into the agar were recorded, I wonder whether the finding of variable susceptibility to bacteria is an intended/unintended consequence of avoidance or attraction of the ants to the agar block that has nothing to do with the existence of antimicrobial surface substances. I wonder whether the authors have considered that some ants will use the agar block as a food source and therefore tunnel into it and through this orally acquire an infection with the bacteria present on the agar block. Unfortunately, the observations on tunnelling are not reported or discussed.
2.2 In their second experiment the authors measured the antimicrobial activity of surface extracts of the ants. While this or similar methods have been used in previous studies to investigate a relationship between antimicrobial strength and sociality (Turnbull et al. 2011) or sociality and group size (Penick et al. 2018) I am highly sceptical that this method to assess the antimicrobial activity of surface substances will give valid insights for the following reasons. 1) This method likely extracts substances from exocrine glands all over the body of the ants and not just substances that are present on the cuticle all the time. Most of these glands will contain compounds that at some concentrations will show antimicrobial activity, especially defensive glands such as the poison gland. An antimicrobial activity of a substances at certain concentrations is hardly surprising, as the dose makes the poison (alleged statement of Paracelsius). What is however completely ignored with this method is whether ants use antimicrobial compounds from these glands for their antimicrobial activity, i.e. actively use them to fight microbes. Without knowing this the antimicrobial activity of compounds, gland or surface extracts is unfortunately only of pharmacological but not really of biological interest. 2) Depending on the solvent used for extraction the method will give varying results as it will extract different compounds from the same gland or compounds form different glands. While the authors acknowledge this partly in line 306-307, this is and reason 1) mentioned above make it impossible to connect the results of the method to create surface extracts with the biology of the species under question.

3. References:
3.1 I also like the authors aim to test for a possible increase in antimicrobial ability in response to exposure to bacteria. As far as I am aware there is only one study that has investigated this with antimicrobial metapleural gland secretions: Yek et al. (2012; https://doi.org/10.1098/rspb.2012.1458). Yet this reference is unfortunately missing in the reference list.
3.2 Line 48: The reference to Cremer et al. (2007) is placed behind part of the sentences talking about antibiotics, suggesting that this reference mainly deals with antibiotics, which is clearly not the case.
3.3 Line 140-142: Here the authors write that reduced carbohydrate levels have been shown to reduce antimicrobial expression and that this was the reason ants were fed sugar water to avoid weakening possible antimicrobial activity. However, with their method of extracting surface substances it is highly unlikely that antimicrobials in the form of antimicrobial peptides were investigated. Instead the authors likely extracted antimicrobials from exocrine secretions with their method.
3.4. Line 328-330: The study of Lucas et al. (2019) although an interesting study itself, is certainly not appropriate to reference strategies used by some ant species in controlling and mitigating the risk of fungal pathogens

Reviewer 3 ·

Basic reporting

This paper introduces many fascinating topics, but it left me scratching my head. I think there could be improvements on clarity, relating the study to the literature, and expanding on relevant details.

I found the conclusions to be a bit vague and confusing. The authors aim to make a connection between the ability to produce antimicrobial compounds and susceptibility to bacteria. There were two ant species (S. invicta and B. chinensis) that were used in both experiments. B. chinensis extracts had relatively low inhibitory effects on E. coli, yet they enjoyed relatively high survivorship when exposed to it (the opposite of what the authors hypothesize). Additionally, the other ant species, S. invicta, had relatively strong inhibitory effects against E. coli, yet suffered higher mortality when exposed (also opposite of hypothesis). This suggests that the antibacterial activity of extracts do not predict protection against the bacteria in the setting they tested. If these results are valid, this is perplexing, but the authors don’t discuss what could be explaining this pattern. The authors still say in the discussion “our results could be an indication of a linkage between an ant species’ susceptibility to bacterial exposure and antimicrobial ability” (lines 290-291) and end the discussion saying “We hypothesize that ants that are susceptible to environmental bacteria are more likely to employ antimicrobials (and perhaps less likely to rely on immune responses)” (lines 332-334). Seems odd to end with a hypothesis that was not supported by the study. (Also, it would be helpful to have this hypothesis more clearly stated in the introduction.)

The authors call the experimental bacteria “non-pathogenic” but go on to state that the goal of the experiment was to test if those bacteria had harmful effects. I would argue that it isn’t accurate to call something non-pathogenic if its pathogenic properties haven’t yet been tested. I suggest the authors change this descriptor to environmental bacteria, to indicate that they are not ant-specific. I don't think a microbe has to be coevolved with a specific host species to cause disease.

I found there to be little background information about the bacteria, especially Staphylococcus epidermidis. It would be helpful to explain whether or not this is a bacterial species that ants are likely to encounter in nature. In Penick et al. 2018, they say: “Testing inhibition of Staphylococcus bacteria is the standard technique to measure antimicrobial activity of insect compounds, and it is the same approach used for the comparative studies in insect groups described above [11–13,26,27]”. This justification, and a similar one for E. coli, would be helpful to have in the text.

The authors do not explain the reasons for picking these four species, how exactly they differed in antimicrobial activity from Penick et al. 2018, or how they are related. The authors also discuss potential medical applications but don’t relate it to the results, the bacteria used in the study, or the compounds the ants produce. It seems like there is a lot of interesting ideas in there, but none are really flushed out and tied to the study.

Much of the paper is based off a previous paper (Penick et al. 2018). The authors report findings from that paper inaccurately. 40% of the species in that study were ineffective at inhibiting growth (line 76 says 40% were effective). In the Discussion, the authors bring up the fact that they found a result that contradicts this previous study (lines 304-307), which they speculate may be due to different extraction solvent. This could be discussed more and could make a call for repeating Penick’s experiment since it appears it’s possible some of the 40% of species that did not show responses to S. epidermidis may need isopropanol to be extracted.

There were also numerous language and grammatical errors, which I did not comment on here.

Experimental design

200 workers were used for S. invicta instead of 50 like the other species. If they did not use larger petri dishes for S. invicta, density of ants would impact their results by altering likelihood of ingesting the bacteria or experiencing allogrooming. It seems to me this should be discussed, and the results for S. invicta should be plotted and interpreted separately from the other species.

I found it to be problematic that there was no estimation of the number of bacterial cells on each piece of agar that the ants were exposed to. And was the bacterial density or abundance increased in S. invicta’s dishes? I think this information would be important for interpreting the results.

I don’t quite understand the explanation of the unbalanced design in lines 224-227. Are nests different than colonies?

“Control group responses were subtracted from treatment group responses to directly compare treatment effects” (lines 234-235). I'm not sure I understand why the authors did this. How could they compare the treatments to the controls if one was subtracted from the other? The figures are unintuitive – I think it would be easier to just plot the control values alongside the treatments instead of subtracting.

The authors say that the ants in Experiment 2 were “subjected to a sub-lethal period of bacterial exposure” (lines 293-294) but that “dead individuals were removed” (lines 192-193; and they don’t state how many individuals died). The authors then used “the remaining living workers for extraction”. Assuming a different number of individuals died from each species (especially since their first experiment showed differences in mortality), wouldn’t that result in different numbers of workers being used for the extractions? It would be helpful if they included how many workers died in each group if they believe that difference in workers didn’t affect their results.

Validity of the findings

The data and analyses for the second experiment are not provided.

There are areas where confounding variables might be present, which I bring up in the Experimental Design comments (i.e. the number of individuals used for the extractions, the number of bacterial cells administered), which may compromise the validity of the findings if present or unaccounted for.

Additional comments

The authors conducted two studies about ant susceptibility to bacteria. The authors first tested the lethality of two species of bacteria for four ant species. They administered bacteria or control agar to ants in petri dishes and counted the number of dead ants after 24 and 48 hours. They found that two of the four species had higher mortality when exposed to bacteria compared to the control ants. One of these species was only susceptible to one of the bacteria; the other was susceptible to both. This experiment was nice and simple and showed that there is variation in susceptibility to different bacteria in ants. I think this is a great result and opens up many other questions.

The authors then sought to test whether these differences in mortality were reflected by inhibitory cuticular antimicrobial compounds and whether this antimicrobial activity was induced by non-lethal exposure (vs. constitutive). This is a very interesting question and nice to pair with the first experiment and a previous study (Penick et al. 2018). They used two of the four species used in experiment one and exposed them to bacteria or control agar for 6 hours before extracting their cuticular compounds and running antibacterial assays on the same bacteria the ants were exposed to.

However, this paper left me feeling confused. There were lots of ideas brought up in the introduction and discussion, but they were not connected to the goal or findings of this study. Additionally, some of the terminology was confusing (e.g. using “non-pathogenic” to describe a bacterial species that has not yet been tested for pathogenic effects). Lastly, I found the methods to be problematic in a few ways; please see the Experimental Design comments.

---

## Round 0.2 · Minor Revisions

· Academic Editor

Minor Revisions

Thank you for your edited manuscript. The reviewer comments were adequately addressed, and I believe the manuscript is nearly ready for publication.

I have a few minor comments that should be addressed before final acceptance.

In lines 60-80, can you specify whether these extracts/venom etc are effective antibiotics at the natural level on the ant? As it is written now it is unclear whether they are effective at killing bacteria at a biologically relevant level for the ant. The final sentence of the paragraph implies that perhaps these previous trials were not biologically relevant, but please make that clear. Specifically, in line 71 – are those small concentrations biologically relevant? Small concentrations is a relative term and it is unclear what the impact is.

Line 115: Is isolating supposed to be inoculating or exposing?

Line 132: Clarify what DB difco LM Miller agar and media are – if it is something you made please give the ingredients and proportions, if it is something you bought please give the supplier information. Don’t refer to it as LB throughout – refer to it as growth media or broth or similar. “LB media” would be fine though.

Line 148-150: the line highlighted in Reviewer 2 comment 3.3 –fasting the animals between collection and experiment could affect their physiology in general, not just for antimicrobial excretions. The choice is yours but if others read it and question it like reviewer 2 did, an alternative would be to simply write something like “in the lab ants were fed sugar water (citation)”. It would be a good idea to give the sugar water concentration.

Line 204-205: The whole body of the ant was in-tact at the end of the extraction, correct? I would imagine that vortexing for 15s at a high speed could cause the bodies to break apart, so please clarify.

Line 211-212: The added line is confusing. You combined the extract of 10 ants, correct? Not combined the 40 ants and then separated into 4 parts? Because that would not be equivalent to 10 worker ants.

Results, experiment 1: put the model/stats results in the text. Please include effect sizes throughout Experiment 1 and 2 results

It is still unclear why you conducted 2 different post hoc tests. Just pick one and report those results.. The independent variables were time, and treatment – did you also test the interaction of time and treatment? It is unclear.

Lines 255-258 are methods not results, but I think you should only report 1 post hoc test.

Lines 265-268 are methods not results.

Lines 275-288 are methods not results.

Reviewer 1 ·

Basic reporting

no comment

Experimental design

no comment

Validity of the findings

no comment

Additional comments

The authors accepted almost all of my previous suggestions. I am satisfied with the changes and believe that the manuscript can be published.

---

## Round 0.3 · accepted · Accept

· Academic Editor

Accept

Thank you for your thoughtful revisions to this manuscript. It is now ready for publication at PeerJ. It was a pleasure working with you, Congratulations!